# Mitochondrial phylogenomics reveals deep relationships of scarab beetles (Coleoptera, Scarabaeidae)

**Shibao Guo[1]\*, Xingyu Lin[2], Nan Song[2]\***

**1** Xinyang Agriculture and Forestry University, Xinyang, Henan, China, **2** College of Plant Protection, Henan Agricultural University, Zhengzhou, Henan, China

\* sbguo510@163.com (SG); songnan@henau.edu.cn (NS)

## Abstract

In this study, we newly sequenced the complete mitochondrial genomes (mitogenomes) of two phytophagous scarab beetles, and investigated the deep level relationships within Scarabaeidae combined with other published beetle mitogenome sequences. The complete mitogenomes of *Dicronocephalus adamsi* Pascoe (Cetoniinae) and *Amphimallon* sp. (Melolonthinae) are 15,563 bp and 17,433 bp in size, respectively. Both mitogenomes have the typical set of 37 genes (13 protein-coding genes, 22 transfer RNA genes, two ribosomal RNA genes) and an A+T-rich region, with the same gene arrangement found in the majority of beetles. The secondary structures for ribosomal RNA genes (*rrnL* and *rrnS*) were inferred by comparative analysis method. Results from phylogenetic analyses provide support for major lineages and current classification of Scarabaeidae. Amino acid data recovered Scarabaeidae as monophyletic. The Scarabaeidae was split into two clades. One clade contained the subfamilies Scarabaeinae and Aphodiinae. The other major clade contained the subfamilies Dynastinae, Rutelinae, Cetoniinae, Melolonthinae and Sericini. The monophyly of Scarabaeinae, Aphodiinae, Dynastinae, Cetoniinae and Sericini were strongly supported. The Scarabaeinae was the sister group of Aphodiinae. The Cetoniinae was sister to the Dynastinae + Rutelinae clade. The Melolonthinae was a non-monophyletic group. The removal of fast-evolving sites from nucleotide dataset using a pattern sorting method (OV-sorting) supported the family Scarabaeidae as a monophyletic group. At the tribe level, the Onthophagini was non-monophyletic with respect to Oniticellini. Ateuchini was sister to a large clade comprising the tribes Onthophagini, Oniticellini and Onitini. Eurysternini was a sister group of the Phanaeini + Ateuchini clade.

## Introduction

The family Scarabaeidae is one of the largest and most diverse groups of beetles, with more than 30,000 described species worldwide [1–5]. According to feeding preferences, this family was divided into two major groups, namely the saprophagous and phytophagous lineages. Most of the saprophagous scarabs are specialist dung feeders in adults, which are often called

**Data Availability Statement:** The mitogenome sequences of Dicronocephalus adamsi and Amphimallon sp. have been deposited at GenBank (https://www.ncbi.nlm.nih.gov/genbank/) with the accession numbers MZ955033 and ON529251.

**Funding:** This research was funded by the National Natural Science Foundation of China, grant number U1904104, and the Foundation of Central Laboratory of Xinyang Agriculture and Forestry University, grant number FCL202003. The funders had no role in study design, data collection and analysis, decision to publish, or preparation of the manuscript.

**Competing interests:** The authors declare no conflict of interest.

dung beetles and well known to most people due to their relatively large body size, significantly ecological importance and the sacred symbols of *Scarabaeus* in ancient Egypt. Dung beetles are one of the primary utilizers of mammalian dung on Earth. Many species of phytophagous scarab beetles are economically damaging pests because of the high levels of herbivory. For example, the Japanese beetle *Popillia japonica* is broadly invasive and causes millions of dollars of damage worldwide [6–8].

Despite their enormous ecological and economic importance, the phylogenetic relationships within Scarabaeidae remain controversial. The saprophagous scarabs comprise two major subfamily groups: Scarabaeinae and Aphodiinae [9]. The former are predominantly specialist dung feeders, while the latter are generalists feeding on dead or decaying matter like leaf litter and rotting fruits [1, 9]. In addition, some species classified in Hybosoridae, Ochodaeidae, Geotrupidae and other Scarabaeoidea groups were once considered to be the close relatives of Scarabaeinae and Aphodiinae, and all of them constituted the Laparosticti [10]. The phytophagous scarab beetles are traditionally grouped into four main subfamilies (Melolonthinae, Cetoniinae, Dynastinae, and Rutelinae) [11–13]. The phytophagous scarabs are called as Pleurosticti [14]. The Laparosticti and Pleurosticti are often characterized by location of the spiracles [15, 16], but see [17]. The division of Scarabaeoidea into Laparosticti and Pleurosticti is currently considered as untenable. Besides the six main subfamilies mentioned above, a few other much smaller subfamilies were recognized in the Scarabaeidae, such as Aclopinae, Allidiostomatinae, Euchirinae, Orphninae and Pachypodinae [12, 18].

The monophyly of the family Scarabaeidae as a whole and the relationships among the subfamilies have been contentious. Grebennikov and Scholtz (2004) [18] used larval morphological characters to infer the higher-level relationships of Scarabaeoidea. Their results did not support the monophyly of Scarabaeidae. The subfamilies Scarabaeinae and Aphodiinae was recovered as separate clades [18]. In contrast, the molecular phylogenetic study using the 18S rDNA and 28S rDNA sequence data found strong support for the family Scarabaeidae [12]. The monophyly of the dung beetle clade (Scarabaeinae + Aphodiinae) was also supported [12]. A phylogenetic analysis of the Coleoptera using 95 protein-coding genes also supported the monophyly of Scarabaeidae [19]. The phylogenetic analyses based on the mitochondrial genome (mitogenome) sequences consistently recovered a monophyletic Scarabaeidae [3, 20], though with the limited sampling of Scarabaeoidea. However, a phylogenetic analysis of Scarabaeoidea based on four DNA markers (the 18S rDNA and 28S rDNA genes, and the mitochondrial *cox1* and *rrnL* genes) recovered a non-monophyletic Scarabaeidae [21]. Besides the question of Scarabaeidae monophyly, the subfamilies Aphodiinae and Melolonthinae were also questioned in the previous studies [3, 12]. In fact, the classification for the Scarabaeidae differs between systematists and regions. For example, Ahrens often recognized Sericini to be a tribe of Melolonthinae (e.g., [13, 22, 23]). The tribe Sericini as a member of Melolonthinae was also supported by the study based on the 18S and 28S rDNA sequence data [12]. However, Ahrens et al (2014) recovered Melolonthinae to be a paraphyletic assemblage, with Sericini being a separate clade [21]. Several recent phylogenetic analyses based on mitogenomes often recovered sericine chafers outside Melolonthinae, and tended to support the subfamily status of Sericinae in Scarabaeidae [3, 5, 24].

The main aims of the present study are threefold: (i) to test the monophyly of Scarabaeidae; (ii) to elucidate the phylogenetic relationships among the subfamilies within Scarabaeidae; (iii) to explore the utility of the mitogenome sequence data for these phylogenetic relationships. To perform this study, we sequenced two new mitogenomes from the Cetoniinae (*Dicronocephalus adamsi* Pascoe, 1863) and the Melolonthinae (*Amphimallon* sp.). Combined with the published mitogenomes of Scarabaeoidea construct the most comprehensive datasets of the scarab

mitogenome sequences to date. DNA sequence data were analyzed under various inference methods and different data recoding schemes.

## Material and methods

### Ethics statement

No specific permits were required for the insect specimens collected for this study. These specimens were collected on the field of Zhengzhou. The field studies did not involve endangered or protected species. The two insect species sequenced are all common beetle species in China and are not included in the "List of Protected Animals in China".

### Taxonomic sampling

To evaluate the monophyly of Scarabaeidae and the phylogenetic relationships within this family, we chose a set of ingroup and outgroup species satisfying the following conditions: (i) inclusion of members of several major subfamilies, which ensured covering the major clades of the dung scarabs and the phytophagous scarabs; (ii) inclusion of major families within the superfamily Scarabaeoidea; (iii) choosing the Hydrophilidae to root the tree. A total of 118 ingroup species were included: 70 species from Scarabaeinae, three from Aphodiinae, ten from Dynastinae, six from Rutelinae, 12 from Cetoniinae, 14 from Melolonthinae, and three from Sericini (or Sericinae) (S1 Table). Outgroup taxa for Scarabaeidae belonged to the closely-related families Trogidae (two species), Glaresidae (one species), Geotrupidae (two species), Lucanidae (four species), Hybosoridae (three species), and Passalidae (three species). The more distantly related Hydrophilidae (14 species) were also used to be the non- Scarabaeoidea outgroups. In this study, the classification within Scarabaeoidea followed those of Smith (2006) [25].

Three mitogenome sequences downloaded from the public database GenBank (Scarabaeidae sp. BMNH 1274750, Scarabaeidae sp. BMNH 1274752, Scarabaeidae sp. BMNH 1274753). Three sequences were originally classified at the family level. We re-conducted molecular identification using the mitochondrial *cox1* gene. In addition, combined with the resulting trees inferred in the present study, we assigned Scarabaeidae sp. BMNH 1274752 to the tribe Canthonini (Scarabaeinae) and both Scarabaeidae sp. BMNH 1274750 and Scarabaeidae sp. BMNH 1274753 to the tribe Ateuchini (Scarabaeinae).

### Sequencing of complete mitogenomes

**Specimen collection and DNA extraction.** The adult specimens of *D. adamsi* and *Amphimallon* sp. were collected at Zhengzhou, Henan province, China, in July 2019 and 2020, respectively. The specimens were preserved in absolute ethanol and stored at -80°C until DNA extraction. Total genomic DNA was extracted from leg muscle tissue using the TIANamp Genomic DNA Kit (TIANGEN BIOTECH CO., LTD), following the manufacturer's protocol. After DNA extraction, the specimen parts have been deposited in Entomological Museum of Henan Agricultural University (voucher numbers: *D. adamsi*, EMHAU-2022-Zz190706; *Amphimallon* sp., EMHAU-2022-Zz210705). Both specimens in the museum are available for subsequent verification.

**Next generation sequencing.** The extracts were delivered to BGI-Shenzhen, China for library preparation and sequencing. Libraries for sequencing were prepared from cDNA, which was sheared to produce 300–4000 bp double stranded fragments. Sequencing was performed by using the BGISEQ-500 platform, with the strategy of 100 paired-end. Low quality sequencing reads were trimmed by using NGS QC Toolkit [26]. The remaining clean data

(15,947,554 reads for *D. adamsi* and 21,808,396 reads for *Amphimallon* sp.) were then assembled into contigs.

## Mitogenomes assembly, annotation and analysis

Clean reads were used for *de novo* assembly of mitgnoemes of using GetOrganelle v1.7.5.2 [27]. We used the GetOrganelle animal database (-F animal_mt) to identify, filter and assemble target-associated reads.

The mitogenome contigs were initially annotated in the MITOS webserver [28], under the Metazoan RefSeq 63 set and default settings. The gene boundaries of protein-coding genes were refined by alignment with closely related scarab species. Transfer RNA (tRNA) genes were determined and the corresponding secondary structures were inferred in MITOS web [28]. Secondary structures for *rrnL* and *rrnS* genes were predicted by the comparative method, with reference to the scarab beetle *Popillia mutans* [3]. The new mitogenome sequences of *D. adamsi* and *Amphimallon* sp. have been deposited at GenBank, with the accession number of MZ955033 and ON529251.

Nucleotide compositions of the mitogenome sequences were calculated using MEGA X [29]. The strand compositional bias was examined by using AT and GC-skew values calculated by the formula: AT-skew = (A-T)/(A+T) and GC-skew = (G-C)/(G+C) [30].

## Multiple sequence alignment

Sequence alignments of protein-coding genes were accomplished using TranslatorX [31], with the MAFFT algorithm [32]. The poorly aligned regions and gaps were removed by trimAl v1.4 [33], with the "-automated1 heuristic". Ribosomal and transfer RNA genes were aligned individually in MAFFT version 7 [32], with the iterative refinement option of E-INS-i. The resulting alignments were also trimmed using trimAl v1.4 [33], with the above-mentioned parameter settings. Alignments were concatenated with FASconCAT-G_v1.04 [34] to compile the following three types of datasets: 1) PCG_nt, the concatenated nucleotide sequences including 13 protein-coding genes; 2) PCG_aa, the concatenated amino acid sequences including 13 protein-coding genes; 3) PCGRNA, the concatenated nucleotide sequences including 13 protein-coding genes, two rRNA genes and 22 tRNA genes; 4) PCG_nt12, the concatenated nucleotide sequences of 13 protein-coding genes with the third codon positions removed; and 5) PCG12RNA, the concatenated nucleotide sequences of PCG_nt12, two rRNA genes and 22 tRNA genes.

## Phylogenetic analyses

**Tree searches.** We employed both Maximum likelihood (ML) and Bayesian inference (BI) methods to conduct phylogenetic tree reconstructions. Model selection was conducted in ModelFinder [35] for every concatenated matrix used herein, with the alignment partitioned by gene.

ML trees were reconstructed from each of the matrices with IQ-TREE 1.6.12 [36] using the best-scoring substitution model for each gene partition (S2 Table) as selected with ModelFinder [35]. Support for trees was assessed with 10,000 ultrafast bootstrap (BS) replicates [37].

Bayesian tree searches were performed with PhyloBayes-MPI 1.8 [38, 39]. For each dataset, we conducted two independent runs. The CAT-GTR model was used in the analyses of nucleotide alignments, while the CAT-mtZOA model was used in the analysis of amino acid alignment. Each run involved two chains, with a total length of 10,000 cycles. Convergence between chains was evaluated by examining the difference in frequency for all their bipartitions (maxdiff < 0.3). The consensus tree was built from the two runs with the *bpcomp* program

discarding the first 1,000 trees as burn-in. Branch support was assessed by the clade posterior probabilities (PP).

We tested how slow and fast evolving sites could affect phylogenetic discordance. To do so, trees were generated from data matrices produced using the OV-sorting method [40]. This method ranks the originally concatenated alignment from the most variable sites to the most conserved sites based on the measurement of "observed variability" (OV) of each alignment position. Previous studies have suggested that accuracy of phylogenetic inference can be improved by the removal of the most variable sites from datasets using the OV-sorting method [41–43]. A series of shortened alignments were created by sequentially removing fast-evolving sites in 250 increments from the 13,858 sites of the PCGRNA dataset. The OV-sorted datasets were subjected to unpartitioned ML analyses, with the parameter settings as described above.

**Hypothesis testing.**   The specific phylogenetic hypothesis, namely the relationships of major clades of Scarabaeidae, was further assessed by means of the four-cluster likelihood-mapping (FcLM) approach implemented in IQ-TREE 1.6.12 [36]. For the three datasets (PCG_nt, PCG_aa and PCGRNA), we used the same partition schemes and the corresponding models selected during the IQ-TREE tree search.

## Results

### Sequencing of mitochondrial genomes

Genome sequencing produced 15,947,554 and 21,808,396 paired-end reads for *D. adamsi* and *Amphimallon* sp., respectively. The approximately 3.53% and 0.13% of the reads resembled mitochondrial sequences. The 15,563 bp and 17,433 bp contigs were assembled for the mitogenomes of *D. adamsi* and *Amphimallon* sp.. Mean coverage of the two contigs reached 4339-fold and 266-fold.

The mitogneomes of *D. adamsi* and *Amphimallon* sp. have the typical set of 37 mitochondrial genes (13 protein-coding genes, 22 tRNA genes, and two rRNA genes) and a major noncoding region (also called control region or A+T-rich region). Both mitogenomes have the similar gene content and genome organization found in the majority of beetles. Nucleotide composition of the mitogenomes was A+T biased, 74.4% for *D. adamsi* and 73.5% for *Amphimallon* sp., respectively. For the majority strand in two scarab species, the AT-skew is positive (higher A than T content, 0.025 for *D. adamsi* and 0.056 for *Amphimallon* sp.) while the GC-skew is negative (higher C than G content, -0.27 for *D. adamsi* and -0.25 for *Amphimallon* sp.).

The complete set of 22 tRNA genes typical of insect mitogenomes [44] is present in the two mitogenomes. The length of tRNA genes range from 62 bp to 71 bp. All tRNA gene sequences can be folded into the cloverleaf structure, except for *trnS1* in both species and *trnW* in *D. adamsi* (Fig 1 and S1 Fig). The *trnS1* in both species and the *trnW* in *D. adamsi* have an unpaired stretch of 12 nucleotides or 7 nucleotides instead of the DHU arm.

The lengths of *rrnS* genes are 780 bp and 785 bp for *D. adamsi* and *Amphimallon* sp., andthe lengths of *rrnL* genes are 1,303 bp and 1,329 bp, respectively. The secondary structures inferred for the *rrnL* and *rrnS* genes were similar to the secondary structure models proposed for other scarab beetles (e.g., *P. mutans*, [3]). Both *rrnL* genes' secondary structures include five major domains (I, II, IV, V and VI). The sequence spanning domain II and domain IV is very short that leads to a loss of the domain III (Fig 2 and S2 Fig). This pattern is often found in insect *rrnL* gene sequences. Five domains are composed of 41–42 helices. *rrnS* genes have three domains (I, II and III) composed of 27 helices (Fig 3 and S3 Fig).

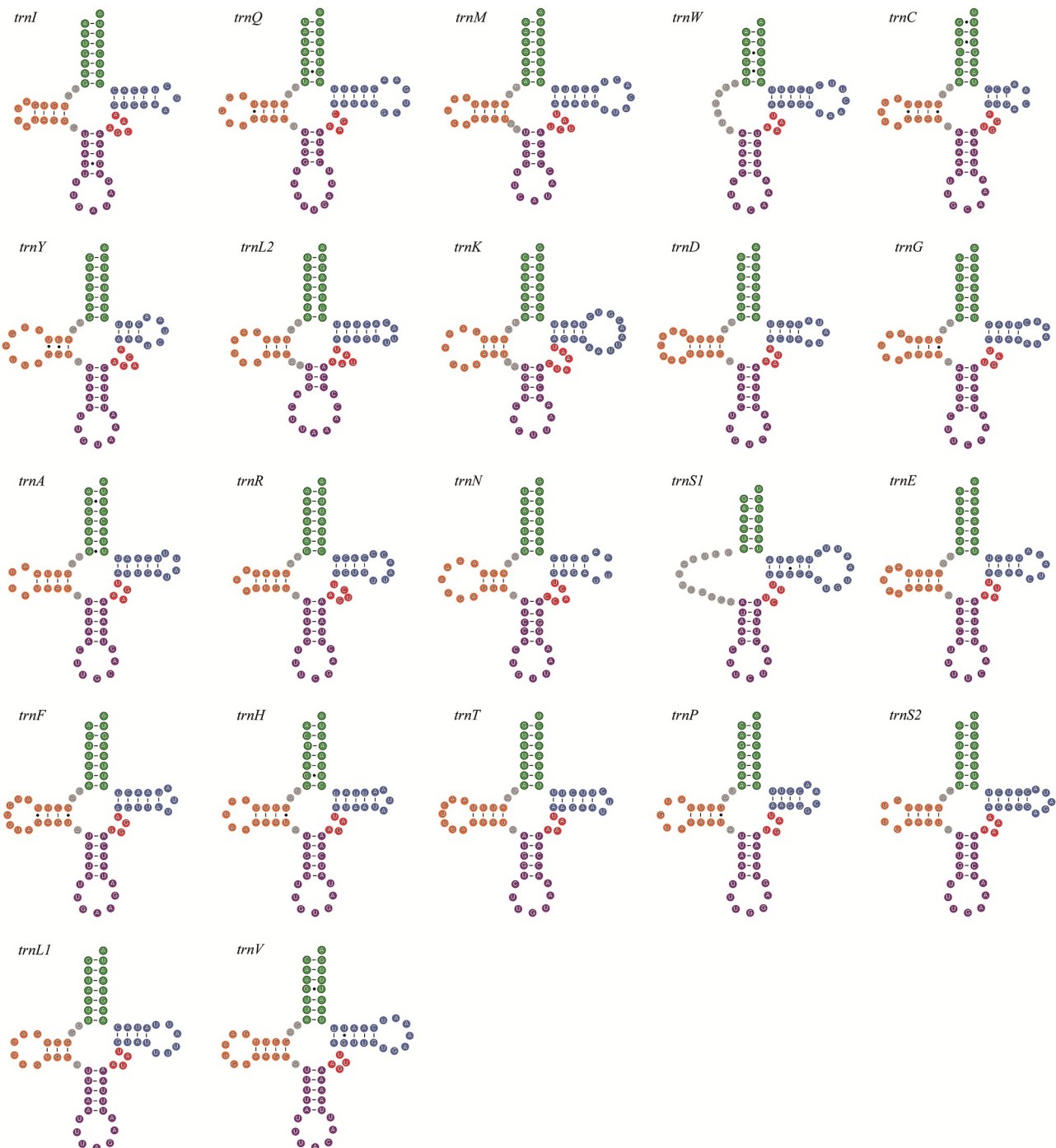

**Fig 1. The secondary structures of tRNA genes inferred for _Dicronocephalus adamsi_.** Dashes in-dicate Watson-Crick base pairing. Dots indicate G-U base pairs.

## Phylogenetic inferences

The same alignment produced the same topology regardless of inference method used. In the ML and BI trees from the amino acid dataset PCG_aa (Figs 4 and 5), the family Scarabaeidae

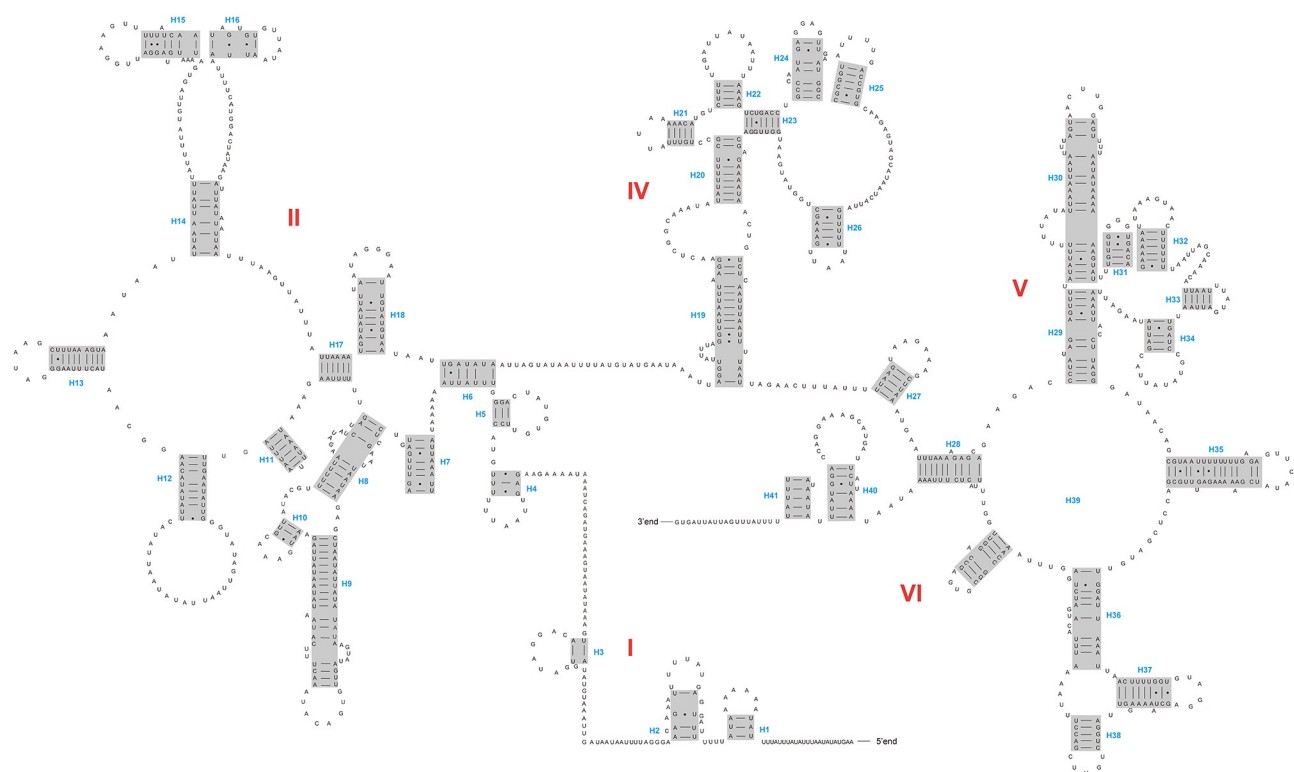

**Fig 2. The secondary structure of *rrnL* inferred for *Dicronocephalus adamsi*.** Dashes indicate Watson-Crick base pairing. Dots indicate G-U base pairs. Red roman numbers denote the domains, and blue numbers denote the helices.

was well supported and split into two major clades. One clade contained the subfamilies Scarabaeinae and Aphodiinae, another included the subfamilies Dynastinae, Rutelinae, Cetoniinae, Melolonthinae and the tribe Sericini. The monophyly of Scarabaeinae, Aphodiinae, Dynastinae, Cetoniinae and Sericini were strongly supported (BS = 100, PP = 1). The node for Rutelinae received the statistically significant support (PP = 0.97) in the BI analysis, but not in the ML analysis (BS = 56). The Melolonthinae was non-monophyletic in both ML and BI trees. In the analyses based on the amino acid dataset, several sister-group relationships were consistently recovered with strong support. The Scarabaeinae was the sister group of Aphodiinae (BS = 100, PP = 1). The Cetoniinae was sister group to the Dynastinae + Rutelinae clade.

The analyses from the nucleotide datasets (PCG_nt and PCGRNA, S4–S7 Figs) yielded tree topologies with the largely identical subfamily relationships in Scarabaeidae. The Scarabaeidae was subdivided into two major clades, which had the same composition and internal arrangement of the subfamilies to those of the analyses from the amino acid dataset. However, the family Scarabaeidae was non-monophyletic with respect to Hybosoridae. The subfamily Rutelinae was recovered as a monophyletic group in the nucleotide analyses. The Melolonthinae was paraphyletic, with the grouping of Cheirotonus + Holotrichia as a separate clade.

Removal of third codon positions did not change tree topological structure in the nucleotide analyses. In the ML tree from the PCG12RNA dataset (S8 Fig), the outgroups (Lucanidae + (Passalidae + Hybosoridae)) was sister to the phytophagous scarab clade, which resulted in a non-monophyletic Scarabaeidae. A similar branching pattern was recovered in the ML analysis of the PCG_nt12 dataset (S9 Fig). In the BI tree from the PCG12RNA dataset (S10 Fig), a sister group relationship between the dung scarab clade (Scarabaeinae + Aphodiinae) and

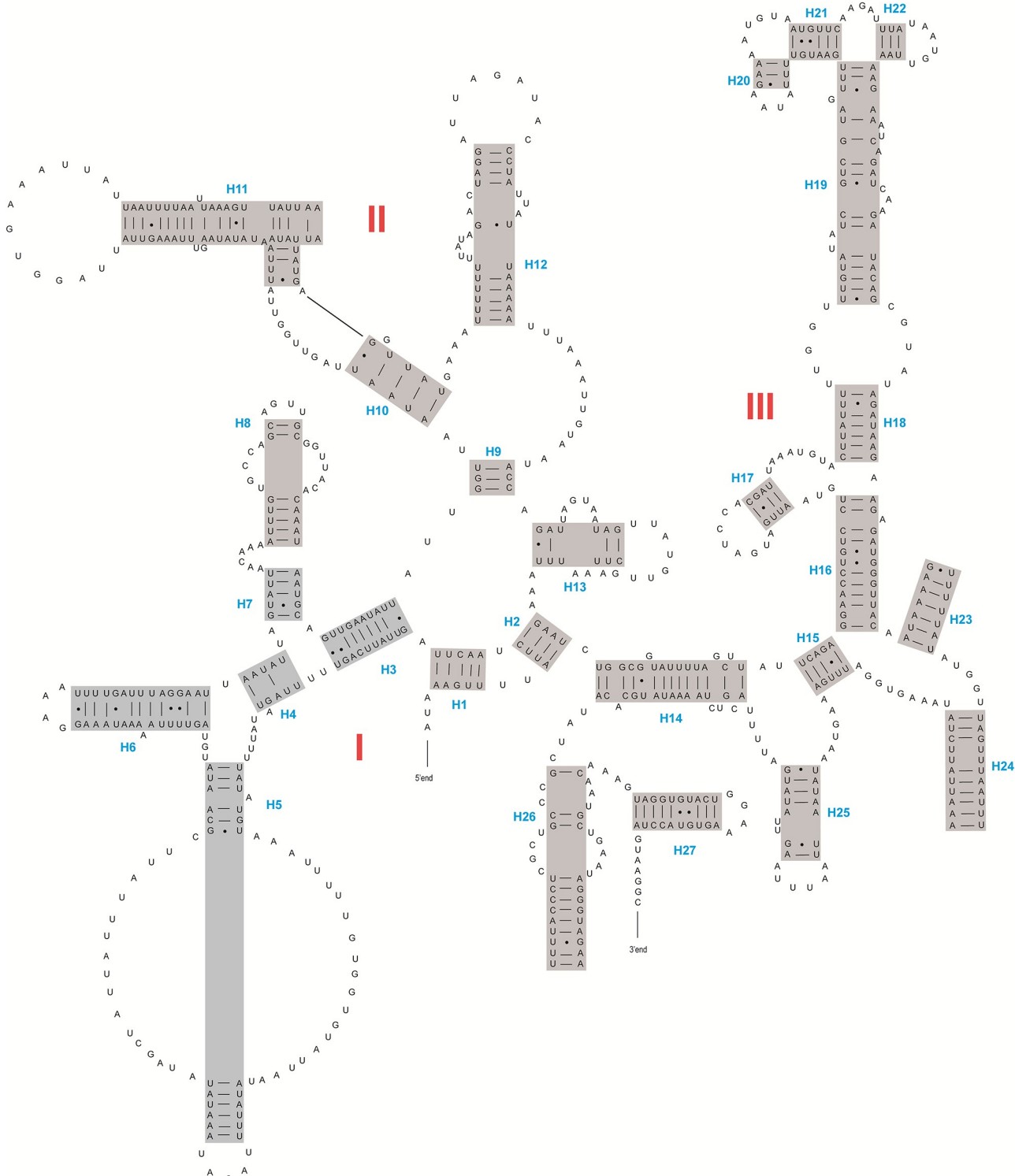

**Fig 3. The secondary structure of *rrnS* inferred for *Dicronocephalus adamsi*.** Dashes indicate Watson-Crick base pairing. Dots indicate G-U base pairs. Red roman numbers denote the domains, and blue numbers denote the helices.

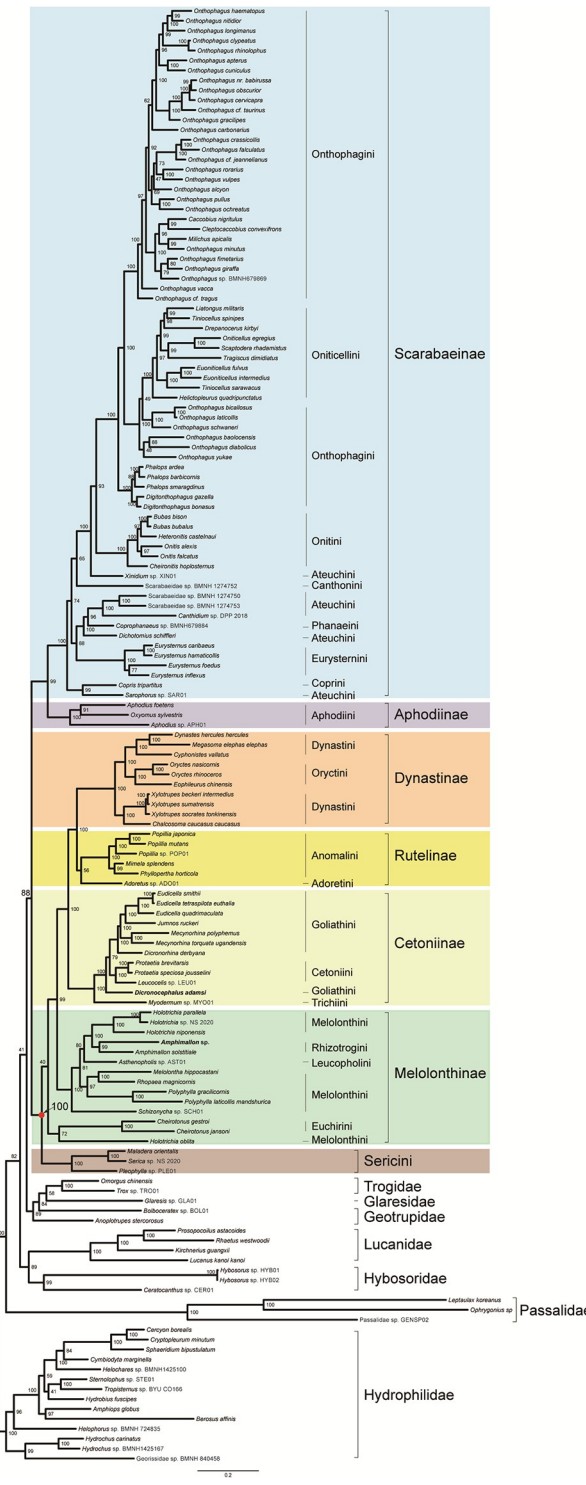

**Fig 4. Phylogenetic tree inferred from the dataset PCG_aa using IQ-TREE under the models selected by ModelFinder.** In the phylogenetic tree, node numbers show the bootstrap support values. Scale bar represents substitutions/site. Bold denotes the newly sequenced species. Colors identify the subfamily groups.

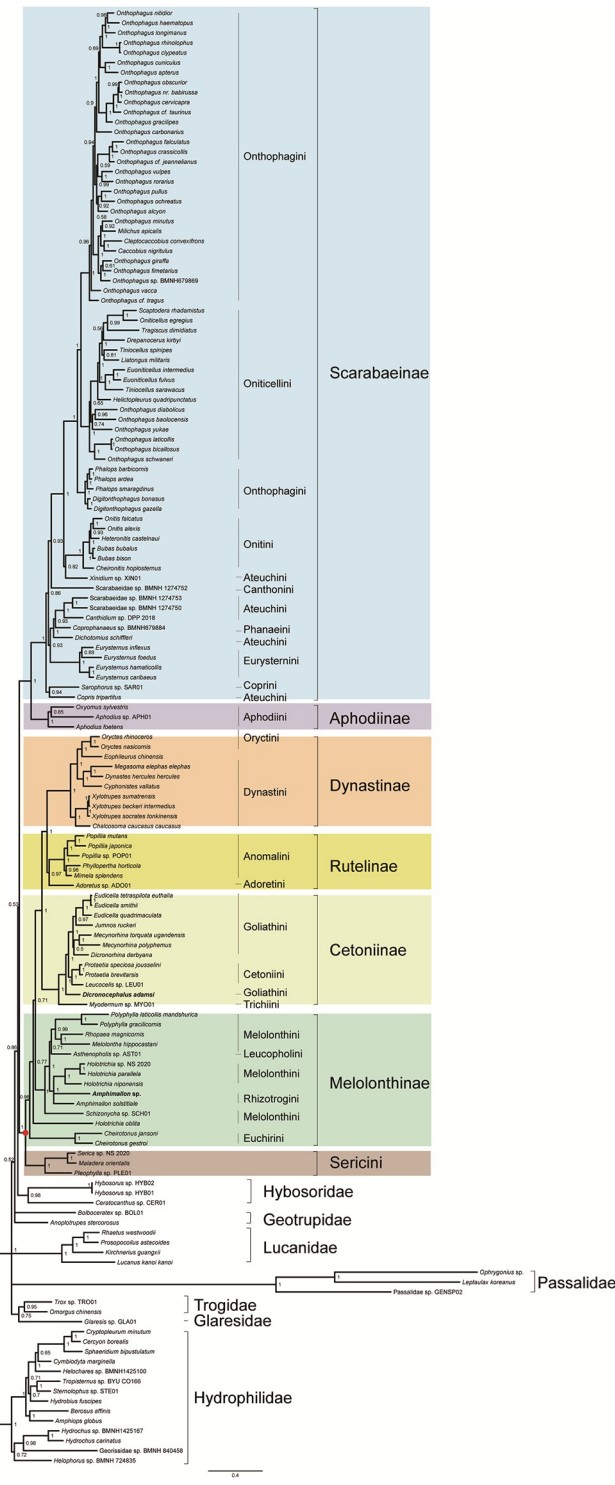

**Fig 5. Phylogenetic tree inferred from the dataset PCG_aa using PhyloBayes-MPI under the model CAT-mtZOA.** In the phylogenetic tree, numbers around nodes indicate posterior proba-bilities. Scale bar represents substitutions/ site. Bold denotes the newly sequenced species. Colors identify the subfamily groups.

Hybosoridae (PP = 0.8) led to the paraphyletic Scarabaeidae. For the PhyloBayes analysis of the dataset PCG_nt12 (S11 Fig), ambiguity represented by polytomies was found in the relationships among the dung scarab clade, the phytophagous scarab clade and the two outgroup families of Passalidae and Hybosoridae. This pattern showed that removing third codon positions resulted in the loss of phylogenetic signal for resolving the deep relationships among the groups. Despite the non-monophyly of Scarabaeidae, terminal relationships among subfamilies of the family were generally stable across analyses of nucleotide datasets.

Sequentially removing the fast-evolving sites from the PCGRNA dataset produced a series of shortened alignments, which contained increasing proportion of conserved sites. As illustrated in Fig 6, the non-phylogenetic Scarabaeidae was retrieved while the 2,000 most variable sites were included in the ML analysis. After these sites were removed, the monophyletic Scarabaeidae was recovered until 5,000 sites were removed. The nodal support value for Scarabaeidae gradually increased from the analysis of the subset containing 11,608 positions to the analysis of the subset containing 9,608 positions. In the ML trees from the subsets further removing the fast-evolving sites in the subsequent three steps, relationships within Scarabaeidae collapsed due to information loss. After this point (subset containing 8,858 positions), the Scarabaeidae was recovered to be non-monophyletic again and the entire relationships on the tree were unresolved.

The FcLM results (Fig 7), showed support for the sister-group relationship between the Scarabaeinae + Aphodiinae clade and the clade including all other scarab subfamilies (58.5%–79.4%), which is congruent with the ML topology inferred from the amino acid dataset. Alternative relationships had poor support: the Scarabaeinae + Aphodiinae clade being sister group to Hybosoridae (14.5%–31.8%), and the clade including Dynastinae, Rutelinae, Cetoniinae, Melolonthinae and Sericini being sister group to Hybosoridae (2.6%–5.7%). The FcML results suggested that the monophyletic Scarabaeidae was indeed better supported by Maximum likelihood.

## Discussion

### The monophyly of Scarabaeidae

Traditionally, based on morphological evidence, the scarab dung beetles and the pleurostict chafers are considered to be sister groups, both of which constitute the monophyletic Scarabaeidae [45–48]. The Scarabaeidae was also recovered in recent molecular studies [19, 20, 49]. However, the Scarabaeidae was not supported by other analyses based on morphological characters [18, 50], multi-locus sequence data [21, 51–53] and single-copy nuclear protein-coding gene data [54]. In the present study, the monophyly of Scarabaeidae was strongly confirmed by ML analysis (BS = 88) based on amino acid data. In the BI analysis based on the same dataset, the Scarabaeidae was recovered, but with only weak support (PP = 0.86).

Analysis based on the nucleotide subsets produced by the OV-sorting methods showed that removal of the most variable sites has a significant effect on the phylogenetic inference of relationships of Scarabaeidae. The subsets including more fast-evolving sites supported Scarabaeidae to be a non-monophyletic group. Whereas, the subsets containing a relatively large number of conserved positions can recover a monophyletic Scarabaeidae.

Overall, we found that all strongly supported nodes in Scarabaeidae were congruent between topologies from the amino acid dataset and the OV-sorted alignments containing more conserved positions. However, the amino acid dataset and the nucleotide dataset with full positions produced conflict result: the former recovered a monophyletic Scarabaeidae, but the latter retrieved a non-monophyletic Scarabaeidae. The possible explanation for the topological difference between phylograms may be base compositional heterogeneity, substitution

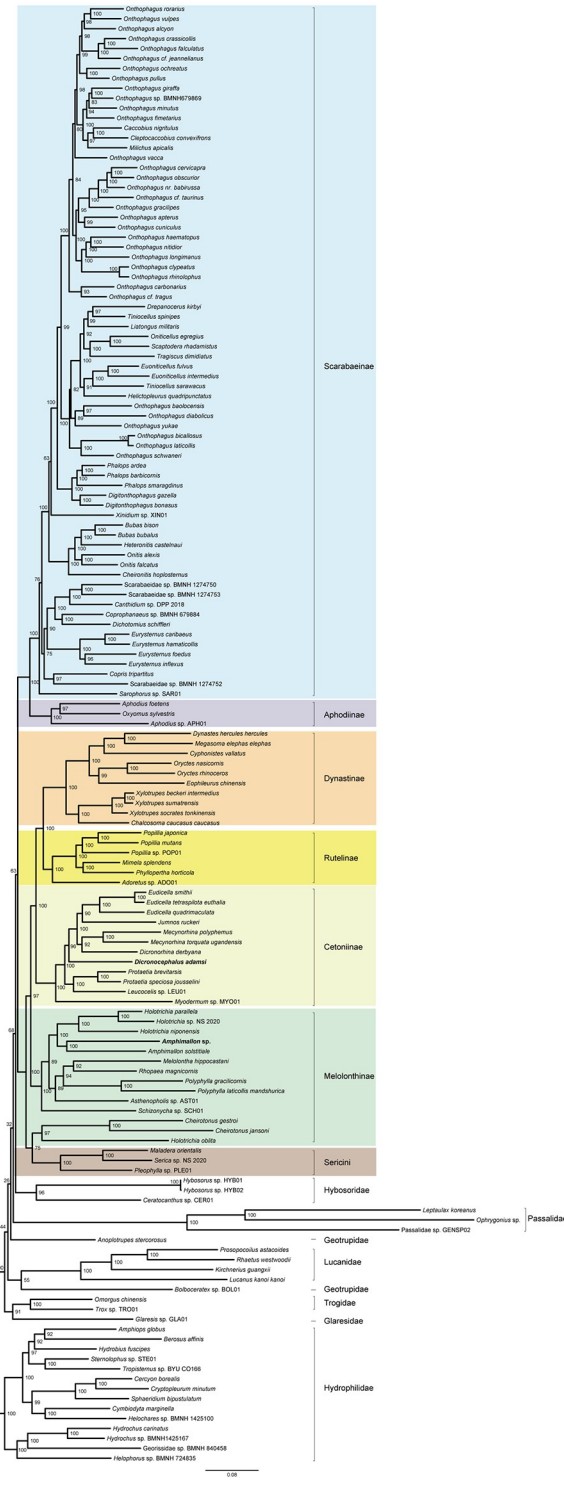

**Fig 6. Phylogenetic tree inferred in IQ-TREE using the subset with 2000 sites removed from the dataset PCGRNA.** Node numbers show the bootstrap support values. Scale bar represents sub-stitutions/site. Colors identify the subfamily groups.

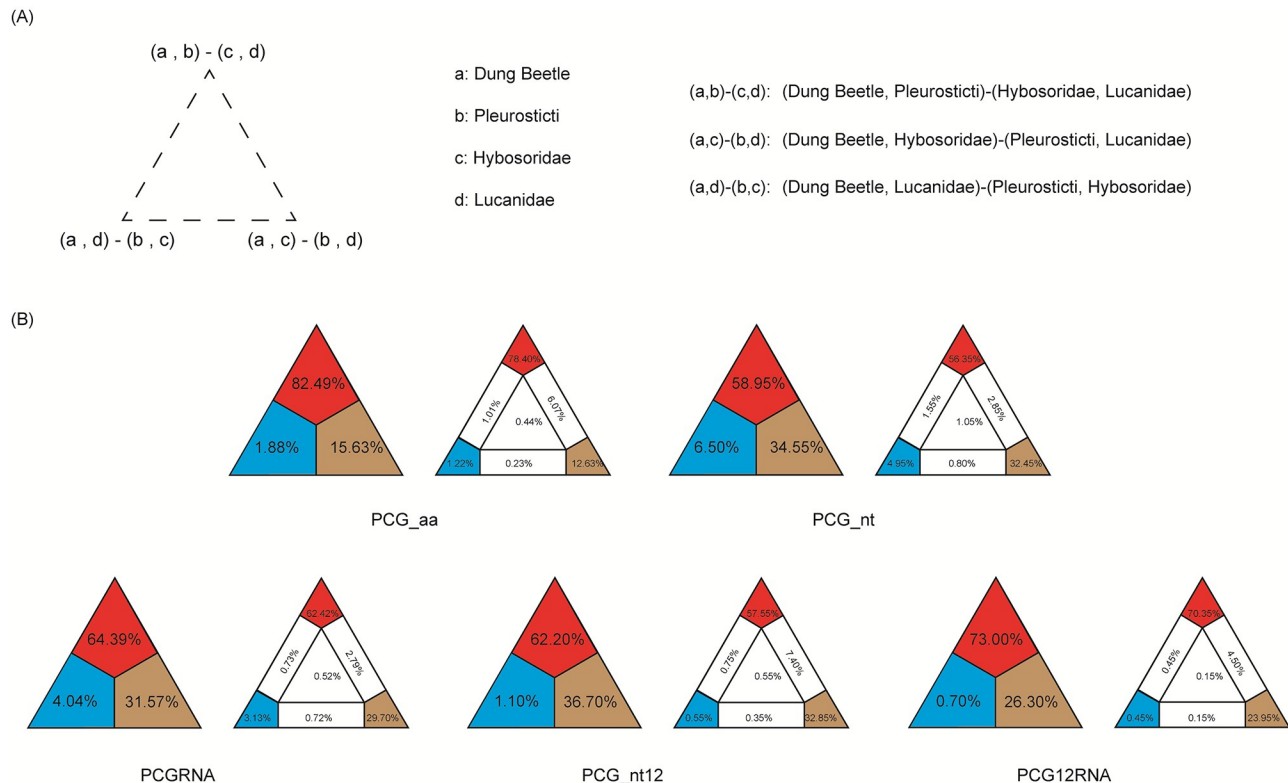

**Fig 7. Results of the Four-cluster likelihood mapping (FcLM) analyses.** (A) Alternative hy-potheses on the relationships of major clades of Scarabaeidae. (B) FcLM results for the concatenated amino acid (PCG_aa) and nucleotide (PCG_nt, PCGRNA, PCG_nt12 and PCG12RNA) alignments.

model misspecification and the potential tree-building artifacts (e.g., long-branch attraction). The FcLM analyses suggested that a sister-group relationship between the dung scarab clade and the phytophagous scarab clade is better supported by ML.

## The interrelationships among subfamilies within Scarabaeidae

Despite the distinct branching pattern on the monophyly of Scarabaeidae inferred from the amino acid dataset and the nucleotide dataset with full positions, both types of data provided the supported phylogenetic hypotheses in splitting Scarabaeidae into two major clades and the interrelationships between subfamilies in each of the clades.

With regard to the dung beetle clade, all analyses recovered the monophyly of Scarabaeinae. Moreover, the sister group relationship of Scarabaeinae and Aphodiinae was strongly supported (BS = 100, PP = 1). This result was highly concordant with previous morphological [46, 50] and molecular studies [12, 20, 51, 53], but contrasted with the studies of Grebennikov and Scholtz (2004) [18] and Hunt et al. (2007) [52]. The Aphodiinae is a large and diverse group, with approximately 3,100 known species [55]. The monophyly of Aphodiinae has been contentious. In a study based on morphological characters, the Aphodiinae was recovered as a paraphyletic group [56]. Molecular evidence also showed Aphodiinae to be a grade, instead of a clade [12]. In a phylogenetic analysis of Scarabaeinae (including fourteen Aphodiinae species as outgroups), the Aphodiinae was recovered as a paraphyletic or non-monophyletic group [57]. The subsequent multiple gene sequence analysis recovered Aphodiinae as non-

monophyletic with respect to the nested Aegialiinae [21]. Our taxon sample of Aphodiinae is very limited due to the mitogenomic data availability. In future researches, more sampling of Aphodiinae and the closely related Aegialiinae are needed to test the monophyly of the group.

The phytophagous scarab clade has long been recognized as a monophyletic group. The earlier studies based on the morphological characters provided strong support for the clade [46, 48, 58]. This clade was further supported by the subsequent analyses based on molecular data [11, 20, 21, 51]. However, the relationships among the subfamilies within the clade remained unstable. In our analyses based on the mitogenomic data, the monophyly of the phytophagous scarab clade was strongly supported (BS = 100, PP = 1). Moreover, we recovered a well-resolved topology with several strongly supported subfamily groups within the clade. The Sericini were unequivocally recovered as sister to the remaining pleurosticts. A sister group relationship between Dynastinae and Rutelinae was confirmed. This arrangement was in accordance with the previous morphological [46, 48, 59] and molecular analyses [11, 24]. Both Dynastinae and Rutelinae formed the sister group of Cetoniinae. This branching order was consistent with that in the study of Ahrens et al. (2014) [21].

In this study, only six mitogenome sequences were included for the Rutelinae due to the data availability of the subfamily. These sequences are from four genera classified in two tribes: the *Adoretus* of Adoretini, and the *Mimela*, *Phyllopertha* and *Popillia* of Anomalini. All analyses clustered the six Rutelinae species in a clade, with Adoretini as a sister group of Anomalini. However, in some analyses (ML PCG_aa, ML PCG_nt, BI PCG_nt, BI PCGRNA, and BI PCG_nt12), there was weak support for a Adoretini + Anomalini sister-group relationship (BS < 70, PP < 0.9). In two prior studies of [9, 21], Rutelinae were recovered as a paraphyletic group. Based on the analyses of four DNA markers (*18S*, *28S*, *16S*, and *cox1*), Ahrens et al. (2014) [21] recovered Adoretini as a sister group of a clade composed of Pachydemini and Dynastinae. Gunter et al. (2016) [9] used the different molecular data (*28S*, *16S*, *12S*, and *cox1*) to yield a similar result to Ahrens et al. (2014) [21]. Rutelinae was recovered as a paraphyletic grade, with Adoretini being sister to Dynastinae [21]. The Anoplognathini and the Rutelini + Anomalini clade formed the independent lineages [21], respectively. Compared with the previous studies [9, 21], our taxon sampling for Rutelinae was incomplete. Increased taxon sampling is expected to generate a more conclusive estimate on the monophyly of Rutelinae.

The Melolonthinae was recovered to be a paraphyletic group. In fact, the monophyly of Melolonthinae has been questioned in previous studies. Ahrens et al. (2014) [21] recovered Melolonthinae as a paraphyletic group based on the analysis of multi-locus data. A prior mitogenomic analysis also supported Melolonthinae as a paraphyletic assemblage [3].

## Tribal relationships within Scarabaeinae and implications for nesting behaviors

Dung beetles in Scarabaeinae were once classified into two main groups based on their rolling and tunneling behavior, namely Scarabaeinae and Coprinae [60]. The former group included six tribes Eurysternini, Deltochilini, Eucraniini, Gymnopleurini, Scarabaeini and Sisyphini, and the latter contained six tribes Coprini, Oniticellini, Onitini, Onthophaghini, Ateuchini and Phanaeini. Although this system was widely accepted by the subsequent systematists on the Scarabaeinae, some questions have been raised on the classification of the subfamily. Montreuil (1998) [61] assigned the genera of Dichotomiini to the tribe Coprini and renamed the tribe Dichotomiini as Ateuchini. Scholtz et al (2009) [62] suggested that Dichotomiini may be a polyphyletic group. A molecular phylogenetic analysis recovered the rolling Canthonini and tunnelling Dichotomiini to be polyphyletic [63]. The phylogenetic analysis based on the morphological characters further supported the polyphyly of Dichotomiini [64]. A morphological

study by Philips et al (2004) [65] did not support the monophyly of the Ateuchini. Bouchard (2011) [66] revised the system of Balthasar (1963) [60] to include 11 tribes in Scarabaeinae, of which Eurysternini was considered as a subtribe in Oniticellini.

For the tribal relationships within Scarabaeinae, our results are consistent with the study of Breeschoten et al. (2016) [67]. The Onthophagini with the extensive taxon sampling was a non-monophyletic group. The monophyletic Oniticellini was nested within a large clade composed of all members of Onthophagini. A previous morphological study has proposed to treat Oniticellini as a subgroup of Onthophagini [68]. The molecular phylogenetic analyses always placed Oniticellini within Onthophagini [48, 49]. This study based on the mitogenomic data confirmed the pattern recovered in the previous analyses [48, 49, 67, 68]. In the ML analysis of amino acid data, the Onitini was recovered as a sister group of the clade including Onthophagini and Oniticellini. A similar result was obtained in the phylogeny of previous studies [57, 67–70].

Ateuchini represented by the species of *Xinidium* sp. was sister to a large clade comprising the tribes Onthophagini, Oniticellini and Onitini. The placement of Ateuchini is similar to that in the study of multi-locus sequence data [70]. The remaining species from Ateuchini were distributed in three clades at the base of the tree, which rendered a non-monophyletic Ateuchini. Based on the analysis of adult morphological characters, Montreuil (1998) [61] recognized Ateuchini to be non-monophyletic.

The Phanaeini represented by a single species of *Coprophanaeus* sp. BMNH679884 was placed in an intermediate position between *Dichotomius* and *Canthidium*. This placement of Phanaeini was incongruent with the previous molecular phylogeny inferred by Monaghan et al. (2007) [57]. In the tree of Monaghan et al. (2007), the *Dichotomius* and *Canthidium* were sister groups, both of which were sister to the Phanaeini + Eucraniini clade [57]. Due to the limited availability of mitogenome sequences, no taxon sampling of Eucraniini was included in the present study. Therefore, there is need for further sequence mitogenomes from the Eucraniini to elucidate the relationships among the tribes.

The monophyletic Eurysternini was a sister group of the clade including Phanaeini and Ateuchini. The *Sarophorus* + *Copris* clade was sister to all other Scarabaeinae lineages. The early diverging position of *Sarophorus* was congruent with Mlambo et al. (2015) [70]. However, the sister group relationship between *Sarophorus* and *Copris* was different from that in Mlambo et al. (2015) [70]. In the tree from Mlambo et al. (2015) [70], both genera had a distant relationship.

According to literatures [65, 69], the *Sarophorus* are non-dung-feeding while Coprini are tunneling. The Eurysternini and Canthonini are recognized as rollers. The tribes Dichotomiini, Phanaeini, Onitini, Oniticellini and Onthophagini and the genus *Xinidium* are often considered as tunnellers. An early phylogenetic analysis using morphological characters did not support the hypothesis of the roller/tunneller classification in Scarabaeinae [71]. In the phylogenetic framework estimated by the current mitogenome sequences, rolling and tunneling behaviors were intermixed throughout the Scarabaeinae lineages. This pattern was consistent with the prior studies [65, 69, 70].

## Conclusion

In this study, we newly sequenced two mitogenomes from scarab beetles belonging to Cetoniinae and Melolonthinae. Combined with the published beetle mitogenomes, we constructed the most comprehensive mitogenome sequence datasets to date. Results provide support for major lineages and relationships among subfamilies of Scarabaeidae. The amino acid dataset and the OV-sorted alignments containing more conserved positions supported the family

Scarabaeidae as a monophyletic group. Given the non-monophyly of the family recovered by the remaining analyses, we encourage researchers to further assess this particular phylogenetic hypothesis in future studies by extending the taxon sampling and using other types of molecular data (e.g., the whole genome-sclae data). The sister group relationships between Scarabaeinae and Aphodiinae and between Dynastinae and Rutelinae were confirmed. The Melolonthinae was non-monophyletic. This study supported Sericini as an independent clade that indicated a subfamily rank of this group. At the tribe level, the Onthophagini was non-monophyletic with respect to Oniticellini. The tribal relationships within Scarabaeinae inferred by the mitogenomic data indicated that rolling and tunneling behaviors were intermixed throughout the lineages. These results are expected to provide insights for a natural classification of Scarabaeidae, and allow for future comparative analyses of dung beetles, including their morphology and nesting behaviors.

## Supporting information

**S1 Fig. The secondary structures of tRNA genes inferred for *Amphimallon* sp.**
(EPS)

**S2 Fig. The secondary structure of *rrnL* inferred for *Amphimallon* sp.**
(EPS)

**S3 Fig. The secondary structure of *rrnS* inferred for *Amphimallon* sp.**
(EPS)

**S4 Fig. Phylogenetic tree of relationships within Scarabaeidae based on Maximum likelihood analysis of the dataset PCG_nt.**
(EPS)

**S5 Fig. Phylogenetic tree inferred from the dataset PCG_nt based on Bayesian inference.**
(EPS)

**S6 Fig. Phylogenetic tree of relationships within Scarabaeidae based on Maximum likelihood analysis of the dataset PCGRNA.**
(EPS)

**S7 Fig. Phylogenetic tree inferred from the dataset PCGRNA based on Bayesian inference.**
(EPS)

**S8 Fig. Phylogenetic tree of relationships within Scarabaeidae based on Maximum likelihood analysis of the dataset PCG12RNA.**
(EPS)

**S9 Fig. Phylogenetic tree of relationships within Scarabaeidae based on Maximum likelihood analysis of the dataset PCG_nt12.**
(EPS)

**S10 Fig. Phylogenetic tree inferred from the dataset PCG12RNA based on Bayesian inference.**
(EPS)

**S11 Fig. Phylogenetic tree inferred from the dataset PCG_nt12 based on Bayesian inference.**
(EPS)

**S1 Table. Taxa included in this study.**
(XLSX)

**S2 Table. The partitioning schemes and best-fitting modes selected by ModelFinder for the dataset (A) PCG_nt, (B) PCG_aa, (C) PCGRNA, (D) PCG_nt12 and (E) PCG12RNA.**
(XLSX)

## Author Contributions

**Conceptualization:** Shibao Guo, Nan Song.

**Data curation:** Xingyu Lin, Nan Song.

**Formal analysis:** Xingyu Lin, Nan Song.

**Funding acquisition:** Shibao Guo, Nan Song.

**Investigation:** Xingyu Lin, Nan Song.

**Methodology:** Shibao Guo, Xingyu Lin, Nan Song.

**Project administration:** Shibao Guo, Nan Song.

**Resources:** Shibao Guo, Nan Song.

**Software:** Xingyu Lin, Nan Song.

**Supervision:** Shibao Guo, Nan Song.

**Validation:** Shibao Guo, Xingyu Lin, Nan Song.

**Visualization:** Shibao Guo, Nan Song.

**Writing – original draft:** Nan Song.

**Writing – review & editing:** Shibao Guo, Nan Song.

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
