## [Decision Letter · Decision Letter 0]

6 Sep 2022

PONE-D-22-21573Mitochondrial phylogenomics reveals deep relationships of scarab beetles (Coleoptera, Scarabaeidae)PLOS ONE

Dear Dr. Song,

Thank you for submitting your manuscript to PLOS ONE. After careful consideration, we feel that it has merit but does not fully meet PLOS ONE’s publication criteria as it currently stands. Therefore, we invite you to submit a revised version of the manuscript that addresses the points raised during the review process.

We look forward to receiving your revised manuscript.

Kind regards,

Pankaj Bhardwaj, Ph.D.

Academic Editor

PLOS ONE

Journal Requirements:

"This research was funded by the National Natural Science Foundation of China, grant number U1904104, and the Foundation of Central Laboratory of Xinyang Agriculture and Forestry Uni-versity, grant number FCL202003."

Please state what role the funders took in the study.  If the funders had no role, please state: ""The funders had no role in study design, data collection and analysis, decision to publish, or preparation of the manuscript."" If this statement is not correct you must amend it as needed. 

3. Please include a copy of Table 1 which you refer to in your text on page 11.

4. Please upload a new copy of Figures 1 to 6 as the detail is not clear. Please follow the link for more information:

https://blogs.plos.org/plos/2019/06/looking-good-tips-for-creating-your-plos-figures-graphics/

https://blogs.plos.org/plos/2019/06/looking-good-tips-for-creating-your-plos-figures-graphics/

Reviewers' comments:

Reviewer's Responses to Questions

**Comments to the Author**

1. Is the manuscript technically sound, and do the data support the conclusions?

Reviewer #1: Partly

Reviewer #2: Yes

2. Has the statistical analysis been performed appropriately and rigorously? 

Reviewer #1: Yes

Reviewer #2: Yes

3. Have the authors made all data underlying the findings in their manuscript fully available?

Reviewer #1: Yes

Reviewer #2: Yes

4. Is the manuscript presented in an intelligible fashion and written in standard English?

Reviewer #1: Yes

Reviewer #2: Yes

5. Review Comments to the Author

Reviewer #1: This publication attempts to address the deep relationships of scarab beetles but falls flat. Only 2 new genome sequences are generated that are then aligned with existing data for over 110 other Scarabaeidae and 15 outgroup Scarabaeoidea. Taxon sampling is unbalanced, mostly due to harvesting existing published datasets published mostly for Scarabaeinae (almost 60% of the data set represents the subfamily and almost 2/3rd of this are Onthophagini- these 2 groups represent ~17% or 5% of the diversity of Scarabaeinae respectively).

Key literature including many existing molecular phylogenies that address these questions in part were not cited. Significantly, these include Breeschoten et al. 2016 for which much of the included sequences were published in, phylogenies that address the monophyly of the Scarabaeidae including McKenna et al. 2014 and Gunter et al. 2016, as well as other order level phylogenies that include significant numbers of Scarabaeoidea that address the question and have produced conflicting results. This is important as it provides context to the degree of novelty of this paper. There are have been a large number of phylogenies that have addressed this question in part, some - especially the most recent papers with dense gene or taxon sampling- are relatively well resolved and provide a broader snap-shot of the deep level relationship that this paper attempts to address.

It is extremely surprising to me that the Lucanidae rendered Scarabaeidae para-phyletic in some analysis. It is generally considered one of the earliest branching members of the superfamily as supported by molecular result and the deep fossil record. This surprising result may indicate that the mt-genome phylogeny is either insufficiently sampled at a taxon-level or is inappropriate for resolving relationships. When analyzed by amino acids and removing the 3rd codon results were more congruent with past analyses.

Ultimately, while I think there are elements of this paper that are novel- the vast majority of this is not. The novelty in this paper revolves around the new genomes, including their descriptions of structure, unfortunately the particular species sequenced do not add novel insights in to poorly resolved relationships in Scarabaeidae. Large components of this paper are just a reanalysis of previously published data from other papers. The results and discussion to not add any new insights in to relationships within the family or subfamilies. For example, the entire section on tribal relationships and nesting behaviors is unnecessary as it is a synthesis of other work that has been discussed in detail in other, sometimes uncited, papers. Furthermore, I disagree with the statement on line 364 that it’s sufficiently sampled (beyond the ingroup of Breeshoten’s work) to address internal relationships, as a number of tribes are not includes, are represented by a single taxon or multiple species in the same genus. I suspect the authors are not familiar with much of the scarab literature because other unanswered questions / unconfirmed relationships in the Scarabaeidae, such as the putative paraphyly of Rutelinae, were not discussed.

The methods were sound and analyses conducted following best practices. I would have liked to see a summary of taxon sampling within the text. The figures were neat and informative. Please note that some taxonomic names within the phylogeny are out of date, for example Digonthophaagus bonasus.

Reviewer #2: Guo et al present a good reanalysis of available Scarab mt genomes with the addition of two extra genomes from subfamilies currently somewhat undersampled. The methods are sound and well detailed allowing ready reanalysis. There are some errors of interpretation and underuse of the available literature which should be corrected prior to final acceptance.

1). Coprophagy. Aphodinae and a significant minority of Scarabaeinae aren't strict dung feeders. They are generalist saprovores, consuming dead/rotting plant material of which dung is a specialised dietary type (this is reflected in the fact that few scarabaeine dung beetle consume dung from non-herbivorous vertebrates). These dietary distinctions and their evolution are discussed at length in Gunter et al. 2016 PLoS One 11 e0153570. It is also worth noting that these distinctions exclusively concern adult diet - almost all Scarabs are saprovores as larvae, specialisation between phyto, sapro and coprophagy is only found in adults.

2) Literature comparisons. The taxon sampling used here, while extensive from the mt genome point of view, is restricted relative to the available molecular phylogenies of the family. For example, just 3sp of Aphedinae, 6 Rutelinae (and just 3 genera) etc. The comparisons to prior literature thus need to be more refined that direct equivalence of subfamily or tribes between the present study and past ones. It is necessary to examine if the representatives within each higher taxon are comparable studies, or if the ones included here represent only particular clades included in other studies. For example Rutelinae is paraphyletic in both Gunter 2016 and Ahrens 2014, forming 3 clades and Dynastinae within the Rutelinae. The relationships found in the present study could be completely accurate if only species from 1 of those three clades were included i.e. both subfamilies monophletic, it would just be misleading as Rutelinae is undersampled to test those prior relationships. Similar issues with the Scarabaeinae.

I would like to see a more detailed examination of how well sampled clades are rather than just statements as to monophyly and relationship with other clades.

Additionally, as devote substantial discussion space to tribal relationships in Scarabaeinae, tribe labels should be labelled on your figure for ease of reading. This section could also benefit from more extensive use of recent literature on behavioural evolution as noted above for Coprophagy.

6. PLOS authors have the option to publish the peer review history of their article (what does this mean?). If published, this will include your full peer review and any attached files.

Reviewer #1: No

Reviewer #2: No

---

## [Author Response · Author response to Decision Letter 0]

27 Sep 2022

Responses to comments

Dear Reviewers, 

Thank you for your very careful review of our paper, and for the comments, corrections and suggestions that ensued. A major revision of the paper has been carried out to take all of them into account. And in the process, we believe the paper has been significantly improved.

In the present “Responses to comments”, we sequentially address all of the points raised in each of the comments made by different reviewers.

Reviewer #1:

Q1: Breeschoten et al. 2016 for which much of the included sequences were published in, 

Response: This reference has been cited in Table S1. 

In the old version, we find the references associated with the mitogenome sequences from GenBank. Although much of the included sequences are from the Breeschoten et al. 2016, the references in GenBank are not updated where the state of “JOURNAL” is still “Unpublished”. According to your comment, we read the paper of Breeschoten et al. 2016 and checked the mitogenome sequences included in it. Indeed, much of the included sequences come from this paper. 

Thank you for this comment.

Q2: phylogenies that address the monophyly of the Scarabaeidae including McKenna et al. 2014 and Gunter et al. 2016 especially the most recent papers with dense gene or taxon sampling- are relatively well resolved and provide a broader snap-shot of the deep level relationship that this paper attempts to address.

Response: After careful inspection, we found that McKenna et al. 2014 you mentioned might be the one of McKenna et al. 2015 (please see details in the following). 

McKenna DD, Farrell BD, Caterino MS, Farnum CW, Hawks DC, Maddison DR, et al. Phylogeny and evolution of Staphyliniformia and Scarabaeiformia: forest litter as a stepping-stone for diversification of non-phytophagous beetles. Syst. Entomol. 2015; 40: 35–60. doi: 10.1111/syen.12093

This paper included the content of addressing the monophyly of the Scarabaeidae. This reference has been cited in this version. Page 12 Line 303, and Page 15 Lines 395-396.

In addition, we also read carefully the paper of Gunter et al. 2016 and cited it in this version. Page 3 Lines 52, 55 and Page 14 Lines 356-360.

Q3: It is extremely surprising to me that the Lucanidae rendered Scarabaeidae para-phyletic in some analysis. It is generally considered one of the earliest branching members of the superfamily as supported by molecular result and the deep fossil record. This surprising result may indicate that the mt-genome phylogeny is either insufficiently sampled at a taxon-level or is inappropriate for resolving relationships. When analyzed by amino acids and removing the 3rd codon results were more congruent with past analyses.

Response: I am sorry that this is a mistake in writing. In the preliminary analysis on the datasets with the different taxon sampling, some resulting trees might please Lucanidae nested within the Scarabaeidae. In the present version, there are no trees with Lucanidae nested within the Scarabaeidae. The associated sentence has been deleted.

Q4: Ultimately, while I think there are elements of this paper that are novel- the vast majority of this is not. The novelty in this paper revolves around the new genomes, including their descriptions of structure, unfortunately the particular species sequenced do not add novel insights in to poorly resolved relationships in Scarabaeidae. Large components of this paper are just a reanalysis of previously published data from other papers. The results and discussion to not add any new insights in to relationships within the family or subfamilies. For example, the entire section on tribal relationships and nesting behaviors is unnecessary as it is a synthesis of other work that has been discussed in detail in other, sometimes uncited, papers. 

Response: The paper of Breeshoten et al. (2016) has been cited in this version (Page 15 Lines 390-391, and Page 16 Lines 397-399). We added some discussions on the comparisons between our results and theirs. With regard to the comments from another reviewer, we added the tribe labels in Figures 4-5 and retained this section in the text.

Q5: Furthermore, I disagree with the statement on line 364 that it’s sufficiently sampled (beyond the ingroup of Breeshoten’s work) to address internal relationships, as a number of tribes are not includes, are represented by a single taxon or multiple species in the same genus. I suspect the authors are not familiar with much of the scarab literature because other unanswered questions / unconfirmed relationships in the Scarabaeidae, such as the putative paraphyly of Rutelinae, were not discussed.

Response: In this version, we compared the taxon sampling of Rutelinae with the two prior studies of Ahrens 2014 and Gunter 2016. For the monophyly of Rutelinae, we have added new discussions in Page 14 Lines 350-366.

Q6: I would have liked to see a summary of taxon sampling within the text. The figures were neat and informative. Please note that some taxonomic names within the phylogeny are out of date, for example Digonthophaagus bonasus.

Response: The species name Onthophagus bonasus is from NCBI database (KU739459) https://www.ncbi.nlm.nih.gov/nuccore/KU739459.1/. 

In this version, we have corrected Onthophagus bonasus to Digitonthophagus bonasus throughout the trees.

Reviewer #2:

Q1: 1). Coprophagy. Aphodinae and a significant minority of Scarabaeinae aren't strict dung feeders. They are generalist saprovores, consuming dead/rotting plant material of which dung is a specialised dietary type (this is reflected in the fact that few scarabaeine dung beetle consume dung from non-herbivorous vertebrates). These dietary distinctions and their evolution are discussed at length in Gunter et al. 2016 PLoS One 11 e0153570. It is also worth noting that these distinctions exclusively concern adult diet - almost all Scarabs are saprovores as larvae, specialisation between phyto, sapro and coprophagy is only found in adults.

Response: Thanks for this comment. The corresponding statements have been revised and the paper of Gunter et al. 2016 has been cited in this version. Please see details in Page 3 Lines 42-44, 52-55.

Q2: 2) Literature comparisons. The taxon sampling used here, while extensive from the mt genome point of view, is restricted relative to the available molecular phylogenies of the family. For example, just 3sp of Aphedinae, 6 Rutelinae (and just 3 genera) etc. The comparisons to prior literature thus need to be more refined that direct equivalence of subfamily or tribes between the present study and past ones. 

It is necessary to examine if the representatives within each higher taxon are comparable studies, or if the ones included here represent only particular clades included in other studies. 

For example Rutelinae is paraphyletic in both Gunter 2016 and Ahrens 2014, forming 3 clades and Dynastinae within the Rutelinae. The relationships found in the present study could be completely accurate if only species from 1 of those three clades were included i.e. both subfamilies monophletic, it would just be misleading as Rutelinae is undersampled to test those prior relationships. Similar issues with the Scarabaeinae. 

Response: In this version, we compared the taxon sampling of Rutelinae with the two prior studies of Ahrens 2014 and Gunter 2016. For the monophyly of Rutelinae, we have added new discussions in Page 14 Lines 350-366.

Q3: I would like to see a more detailed examination of how well sampled clades are rather than just statements as to monophyly and relationship with other clades.

Additionally, as devote substantial discussion space to tribal relationships in Scarabaeinae, tribe labels should be labelled on your figure for ease of reading. This section could also benefit from more extensive use of recent literature on behavioural evolution as noted above for Coprophagy.

Response: We compared the taxon sampling of the major clade within Scarabaeidae included in this study to the prior studies, particularly the studies of Breeshoten et al. (2016), Ahrens et al. (2014) and Gunter et al. (2016). We added the discussions on the tribe relationships within Scarabaeinae (Page 15 Lines 390-391, and Page 16 Lines 397-399) and the monophyly of Rutelinae (Page 14 Lines 350-366). In addition, we added the tribe labels in Figures 4-5.

---

## [Decision Letter · Decision Letter 1]

15 Nov 2022

PONE-D-22-21573R1Mitochondrial phylogenomics reveals deep relationships of scarab beetles (Coleoptera, Scarabaeidae)PLOS ONE

Dear Dr. Song,

Thank you for submitting your manuscript to PLOS ONE. After careful consideration, we feel that it has merit but does not fully meet PLOS ONE’s publication criteria as it currently stands. Therefore, we invite you to submit a revised version of the manuscript that addresses the points raised during the review process.

All the reviewers’ comments are appreciated and considered in issuing the decision as per PLOS ONE’s publication criteria (https://journals.plos.org/plosone/s/criteria-for-publication). Reviewer 1 felt that authors have sequenced only two species and the rest of the data was from the public forum. I personally was OK with the methodology as the authors are able to prove their case with no doubts. Both Reviewer 1 and 3 provided comments on the Discussion section, which should be addressed by the authors.

We look forward to receiving your revised manuscript.

Kind regards,

Pankaj Bhardwaj, Ph.D.

Academic Editor

PLOS ONE

Journal Requirements:

Additional Editor Comments (if provided):

Reviewers' comments:

Reviewer's Responses to Questions

**Comments to the Author**

1. If the authors have adequately addressed your comments raised in a previous round of review and you feel that this manuscript is now acceptable for publication, you may indicate that here to bypass the “Comments to the Author” section, enter your conflict of interest statement in the “Confidential to Editor” section, and submit your "Accept" recommendation.

Reviewer #1: All comments have been addressed

Reviewer #2: All comments have been addressed

Reviewer #3: All comments have been addressed

2. Is the manuscript technically sound, and do the data support the conclusions?

Reviewer #1: Partly

Reviewer #2: (No Response)

Reviewer #3: Yes

3. Has the statistical analysis been performed appropriately and rigorously? 

Reviewer #1: Yes

Reviewer #2: (No Response)

Reviewer #3: Yes

4. Have the authors made all data underlying the findings in their manuscript fully available?

Reviewer #1: Yes

Reviewer #2: (No Response)

Reviewer #3: Yes

5. Is the manuscript presented in an intelligible fashion and written in standard English?

Reviewer #1: Yes

Reviewer #2: (No Response)

Reviewer #3: Yes

6. Review Comments to the Author

Reviewer #1: This manuscript has been improved due to the inclusion of relevant literature, and attempt to consider taxon sampling in the discussion.

It would be very helpful to identify the Amphimallon species. Best practices would be to deposit both specimens in a museum as a voucher that is available for subsequent verification.

The discussion of monophyly of Aphodiinae could be improved as many papers recover it as a paraphyletic grade and not monophyletic.

Reviewer #2: (No Response)

Reviewer #3: The conclusion part seems to be incomplete after all the molecular work, as the final inferences does not include the significance and applicability of the work in the present scenario. Also, some evolutionary aspects in form of final lineage and its relatedness with important model organisms should be mentioned. Futuristic approach also should be addressed compliantly in relation with the present research work as designed by the Authors.

7. PLOS authors have the option to publish the peer review history of their article (what does this mean?). If published, this will include your full peer review and any attached files.

Reviewer #1: No

Reviewer #2: No

Reviewer #3: No

---

## [Author Response · Author response to Decision Letter 1]

17 Nov 2022

Response to reviewers

Dear Reviewers, 

Thank you for your very careful review of our paper, and for the comments, corrections and suggestions that ensued. A minor revision of the paper has been carried out to take all of them into account. And in the process, we believe the paper has been significantly improved.

In the present “Responses to comments”, we sequentially address all of the points raised in each of the comments made by different reviewers.

Reviewer #1:

Q1. It would be very helpful to identify the Amphimallon species. Best practices would be to deposit both specimens in a museum as a voucher that is available for subsequent verification.

Response: Because the insect of this species is large, only leg was used for DNA extraction. The remaining specimen with a relatively complete insect body has been deposited in Entomological Museum of Henan Agricultural University (voucher number: Amphimallon sp., EMHAU-2022-Zz210705). The corresponding statements have been added in this version (Page 6 Lines 136-140).

Q2. The discussion of monophyly of Aphodiinae could be improved as many papers recover it as a paraphyletic grade and not monophyletic.

Response: The discussions and references on the monophyly of Aphodiinae have been added in this version (Page 14 Lines 340-350).

Reviewer #3: 

Q1. The conclusion part seems to be incomplete after all the molecular work, as the final inferences does not include the significance and applicability of the work in the present scenario. Also, some evolutionary aspects in form of final lineage and its relatedness with important model organisms should be mentioned. 

Response: The relevant conclusions have been added in the section of conclusion (Page 17 Lines 452-453, and Page 18 Lines 454-456, 462-464).

Q2. Futuristic approach also should be addressed compliantly in relation with the present research work as designed by the Authors.

Response: The associated statements have been added in Page 14 Lines 348-350 and in the section of conclusion (Page 18 Lines 454-456).

---

## [Decision Letter · Decision Letter 2]

28 Nov 2022

Mitochondrial phylogenomics reveals deep relationships of scarab beetles (Coleoptera, Scarabaeidae)

PONE-D-22-21573R2

Dear Dr. Song,

We’re pleased to inform you that your manuscript has been judged scientifically suitable for publication and will be formally accepted for publication once it meets all outstanding technical requirements.

Kind regards,

Pankaj Bhardwaj, Ph.D.

Academic Editor

PLOS ONE

Additional Editor Comments (optional):

Reviewers' comments:

Reviewer's Responses to Questions

**Comments to the Author**

1. If the authors have adequately addressed your comments raised in a previous round of review and you feel that this manuscript is now acceptable for publication, you may indicate that here to bypass the “Comments to the Author” section, enter your conflict of interest statement in the “Confidential to Editor” section, and submit your "Accept" recommendation.

Reviewer #3: All comments have been addressed

2. Is the manuscript technically sound, and do the data support the conclusions?

Reviewer #3: Yes

3. Has the statistical analysis been performed appropriately and rigorously? 

Reviewer #3: Yes

4. Have the authors made all data underlying the findings in their manuscript fully available?

Reviewer #3: Yes

5. Is the manuscript presented in an intelligible fashion and written in standard English?

Reviewer #3: Yes

6. Review Comments to the Author

Reviewer #3: All the necessary comments have been addressed according to the queries, especially the Conclusion part.

7. PLOS authors have the option to publish the peer review history of their article (what does this mean?). If published, this will include your full peer review and any attached files.

Reviewer #3: No

---

## [Editor Report · Acceptance letter]

2 Dec 2022

PONE-D-22-21573R2 

Mitochondrial phylogenomics reveals deep relationships of scarab beetles (Coleoptera, Scarabaeidae) 

Dear Dr. Song:

I'm pleased to inform you that your manuscript has been deemed suitable for publication in PLOS ONE. Congratulations! Your manuscript is now with our production department. 

Kind regards, 

on behalf of

Dr. Pankaj Bhardwaj 

Academic Editor

PLOS ONE